# Pyridinium Salts of Dehydrated Lanthanide Polychlorides

**DOI:** 10.3390/molecules28010283

**Published:** 2022-12-29

**Authors:** Roger E. Cramer, Esteban M. Baca, Timothy J. Boyle

**Affiliations:** 1Department of Chemistry, University of Hawaii—Manoa, 2545 McCarthy Mall, Honolulu, HI 96822, USA; 2Advanced Materials Laboratory, Sandia National Laboratories, 1001 University Boulevard, SE, Albuquerque, NM 87106, USA

**Keywords:** lanthanides, chlorides, hydrate, dehydration, pyridinium

## Abstract

The reaction of lanthanide (Ln) chloride hydrates ([**Ln(H_2_O)*_n_*(Cl)_3_]**) with pyridine (py) yielded a set of dehydrated pyridinium (py-H) Ln-polychloride salts. These species were crystallographically characterized as [[py-H-py][py-H]_2_[LnCl_6_]] (**Ln-6**; Ln = La, Ce, Pr, Nd, Sm, Eu, Gd) or [[py-H]_2_[LnCl_5_(py)]] ((**Ln-5**; Ln = Tb, Dy, Ho, Er, Tm, Yb, Lu). The **Ln-6** metal centers adopt an octahedral (*OC*-6) geometry, binding six Cl ligands. The −3 charge is off-set by two py-H moieties and a di-pyridinium (py-H-py) ion. For the **Ln-5** species, an *OC*-6 anion is formed by the Ln cation binding a single py and five Cl ligands. The remaining −2 charge is offset by two py-H^+^ cations that H-bond to the anion. Significant H-bonding occurs between the various cation/anion moieties inducing the molecular stability. The change in structure from the **Ln-6** to **Ln-5** is believed to be due to the Ln-contraction producing a smaller unit cell, which prevents formation of the py-H-py^+^ cation, leading to the loss of the H-bonding-induced stability. Based on this, it was determined that the **Ln-5** structures only exist when the lattice energy is small. While dehydrated polychloride salts can be produced by simply mixing in pyridine, the final structures adopted result from a delicate balance of cation size, Coulombic charge, and stabilizing H-bonding.

## 1. Introduction

Lanthanide-based materials have found widespread use in numerous everyday applications such as electronic, automobile, audio, and battery purposes. These critical materials are typically isolated from monazite ores using complicated ion exchange processes that generate large amounts of waste and involve extensive labor and energy input to obtain pure materials [1,2,3]. Recycling efforts pertaining to the Ln cations have come to the forefront as a means to recover and reuse the already separated Ln-element. Recycling usually involves the dissolution of the Ln in a strong acid (HNO_3_, H_2_SO_4_ [4], or HX [5] (X = Cl, Br, I)). One product isolated from the reaction of Ln-based materials (Ln(0) or Ln_2_O_3_) using hydrochloric acid (HCl) are the Ln, chloro, hydrates that form either [Ln(m-Cl)(H_2_O)_7_]_2_•4Cl (Ln = La, Ce, Pr) or [LnCl_2_(H_2_O)_6_]•Cl (Ln = Nd to Lu) [3], often erroneously, referred to as LnCl_3_•6H_2_O for simplicity. For this effort, this family of precursors are referred to as **[Ln(H_2_O)*_n_*(Cl)_3_]**. Lanthanide halides (LnX_3_) are ubiquitous starting materials for synthetic efforts since the ease of halide metathesis affords the facile production of other synthons [3]. However, for these reactions it is critically important to utilize anhydrous materials to prevent detrimental side-reactions. Hence, methods to dehydrate the **[Ln(H_2_O)*_n_*(Cl)_3_]** precursors is necessary. Typically, this is achieved from the reaction of the starting material with NH_4_Cl to form the ammonium pentachloride Ln-derivative ([(NH_4_)_2_LnCl_5_]), followed by heat and sublimation to remove NH_3_ and HCl [6].

Previously, we found that the Sc-chloro-hydrate could be easily dehydrated merely by dissolving the hydrate in an appropriate solvent [7]. Due to this success, a similar simple dissolution/solvation approach to displace the bound H_2_O was undertaken using **[Ln(H_2_O)*_n_*(Cl)_3_]** and a wide range of solvents. Of these studies, the pyridine dissolution products proved to be anhydrous, forming polychloro, pyridinium (py-H) products (eq 1). The products were identified by crystallographic studies as either [[py-H-py][py-H]_2_[LnCl_6_]] (**Ln-6**; Ln = La to Gd) or [[py-H]_2_[LnCl_5_(py)]] ((**Ln-5**; Ln = Tb to Lu). Details of the synthesis and final structures isolated are presented.
**[Ln(H_2_O)_*n*_(Cl)_3_] + (xs) py****→ [z[py-H-py][Py-H]_2_[LnX_x_(py)_y_]]**(1)
*x* = 6, *y* = 0, *z* = 1: *Ln* = (*n* = 7) *La*, *Ce*, *Pr*, (*n* = 6) *Nd*, *Sm*, *Eu*, *Gd*
*x* = 5, *y* = 1, *z* = 0, *n* = 6: *Ln* = *Tb*, *Dy*, *Ho*, *Er*, *Tm*, *Yb*, *Lu*

## 2. Results and Discussion

Searching for solvents that could easily dehydrate **[Ln(H_2_O)*_n_*(Cl)_3_]** led to the exploration of pyridine. This selection was based on the ready formation of NH_3_ products during NH_4_Cl dehydration processing indicating the facile displacement of water by amines and the unusual structures noted below using pyridine.

A number of Ln-polychloride py-H salts have been reported, including [py-H]_2_[CeCl_6_], [py-H]_2_[LnCl_5_(py)] (where Ln = Eu, Er, Yb) [8]or [py-H][YbCl_4_(py)_2_]•py [8]. Information pertaining to the reported Ce(IV) hexachloride is limited as there are no coordinates supplied and the early issues of the journal are not readily available. The previously reported pentachlorides were prepared by heating the anhydrous LnCl_3_ in diacetone alcohol (DAA) and pyridine for 9 h at 100 °C. The products are octahedrally (*OC*-6) bound Ln cations with 5 inner-sphere Cl and a bound py. The 2-charge is balanced by two lattice H-py cations. The long-range order of the crystal results from H-bonding between Cl atoms and the nearest neighbors H-py. For this report, a study of the products from the simple reaction of **[Ln(H_2_O)*_n_*(Cl)_3_]** heated in a solution of py was undertaken.

### 2.1. Synthesis

Initial efforts focused on the [Ce(m-Cl)(H_2_O)_7_]_2_•4Cl precursor. The sample was slurried in pyridine, centrifuged and the soluble fraction allowed to slowly evaporate. From this ‘wash’, a mixed ligand complex was isolated as [py-H]_2_[Ce(Cl)_2_(H_2_O)_6_][Cl]_3_(H_2_O) (**Ce-H_2_O/py-H)**. Figure 1 shows the structure of **Ce-H_2_O/py-H**. This unusual eight-coordinated monomeric complex reveals that one of the Cl ions moved from the outer-sphere to the inner-sphere while three other Cl atoms remained in the outer-sphere and are charge balanced by py-H moieties. These changes, reduce the dinuclear starting material to a monomer. This is believed to be the first step in the dehydration that was observed below. The quality of the crystal structure solution of **Ce-H_2_O/py-H** is high enough to verify connectivity but not sufficient for publication due to low data parameter ratio but is included here (see Table 1) for completeness. Interestingly, **Ce-H_2_O/py-H** does not show any loss of water; thus, additional studies were undertaken to induce dehydration.

Further studies using the early, lighter **[Ln(H_2_O)*_n_*(Cl)_3_]** precursors were initiated and various **[Ln(H_2_O)*_n_*(Cl)_3_]** were stirred in pyridine overnight (12 h). Instead of a clear solution a white slurry formed. In order to induce maximum substitution, the mixture was heated to a boil, and while hot, the mixture was poured through a glass frit. The clear mother liquor was allowed to sit, yielding crystals in all instances. Based on the isolation, the formation of pyridinium ions was expected with a change in inner and outer-sphere Cl ions but the extent of dehydration was surprising.

### 2.2. Characterization

FTIR data were collected since the Ln-Cl interaction would not complicate the spectrum and the presence of either water or pyridine could be readily observed. The FTIR spectral data for all of the compounds reported are available in the Supporting Information (see Appendix A). The loss of the broad -OH stretch around 3000 cm^−1^ for all of the compounds indicated the products were successfully dehydrated. There were also stretches present above 2461 cm^−1^ consistent with a N-H moiety. These would include the five stretches around 1633, 1599, 1531, 1487, and 1445 cm^−1^ that were previously associated with the nuclear variations consistent with displacement of the pyridine ring. Additionally, the 767 and 700 cm^−1^ stretches observed in the spectra are associated with the C-H and C-C bends of the pyridine rings. There is an additional stretch located around 600 cm^−1^ for the light Ln (La to Gd) and then a shift to 623 cm^−1^ for the heavy Ln (Tb-Lu), which may represent the La-Cl interaction. These assignments are consistent with the py-H salts of other metal halides presented by Cook [9].

#### 2.2.1. Crystal Structures

As the remainder of analytical inspections proved fruitless due to the paramagnetic nature of the cations, the presence of halide ligands, and the inconsistencies (volatility, additional trapped or premature loss) of solvated species, crystal structure determination was primarily used to identify the structures of the final products. With the isolation of the crystal structures (*vide infra)*, powder XRD analyses on each powder was undertaken and compared with the calculated patterns. These are shown in the Supporting Information. In general, the bulk powder experimental patters for **Ln-6** and **Ln-5** samples appear to be in agreement with the larger peaks associated with the calculated patterns; however, the substantial other peaks present indicate potential other products and organic species may be present in the final products. It is of note, that numerous crystals were studied for each cation with a consistent isolation of either the **Ln-6** or **Ln-5** observed for each sample.

Three structures and two compositions were observed in this study: [[py-H-py][py-H]_2_[LnCl_6_]] (**Ln-6**: Ln = La and Ce—crystallize in the orthorhombic space group P2_1_2_1_2_1_; **Ln** = Pr, Nd, Sm, Eu, and Gd—crystallize in the monoclinic space group P2_1_/c) and [[py-H]_2_[LnCl_5_(py)]] (**Ln-5**; Ln = Tb, Dy, Ho, Er, Tm, Yb, and Lu—crystallize in the orthorhombic space group Pna2_1_). Figure 2 and Figure 3 show the representative structure plots of the **Ln-6** [[py-H-py][py-H]_2_[EuCl_6_]] (**Eu-6**) and the **Ln-5** [[py-H]_2_[LuCl_5_(py)]] (**Lu-5**) structures observed in this work. Table 1 lists the data collection parameters for all compounds. Table 2 and Table 3 list the average metrical data for the **Ln-6** and **Ln-5** compounds, respectively.

Since any changes in structural properties within the Ln series are expected to be very small due to the Ln-contraction, the three structure types are also expected to have very similar stabilities. In each case (i.e., **Ln-6** and **Ln-5**) an octahedral LnCl_6_ or LnCl_5_(py) polyanion is surrounded by a shell of py-H and/or py-H-py cations. The py-H cations H-bond to Cl ligands coupled with extensive C-H•••Cl H-bonding, which holds the lattice together for crystal formation. These interactions are shown in the packing diagrams for **Ln-6** and **Ln-5** (Figure 1 and Figure 2b, respectively). For the **Ln-6** species, the compounds are distorted from an ideal *OC*-6 geometry by H-bonds formed between the Cl ligands and the H-py cations. The *trans*-Cl angles range from 168.1 to 178.6°, with an average of 176.6°; however, the *trans*-Cl atoms that are independently bound to a H-py, are at the low end of this range. The N_py_-H•••Cl angles range from 152.1 to 175.6° (av 165.1°). For the **Ln-5**, a distorted *OC*-6 geometry results from the binding of five Cl and one py ligand. The *trans* Cl angles span the relatively small range of 174.9 to 177.6° (av. 176.6); the *trans* Cl---N av 177.3 from a range of 177.1 to 178.0°. The N_py_-H---Cl av 177.4° with a range of 177.1 to 178.0°. The average Ln-Cl distances for **Ln-6** (av 2.73 Å) are slightly longer than that of **Ln-5** (av 2.61 Å) compounds. This also results in longer Cl---Cl distances for **Ln-6** vs. **Ln-5** (av) 3.85 vs. 3.69 Å, respectively. The Ln-N distance for **Ln-5** (av 2.49 Å) are in agreement with literature values [10]. However, it is not readily apparent why the structure changes from **Ln-6** to **Ln-5** based on the metrical data alone.

The lanthanides are often claimed to have uniform chemistry, with the assumption that one structure represents the entire series. This is often based on lore or the small handful of examples where the entire series has been studied [11]. As noted for this series, the structure changes from **Gd-6** to **Tb-5**, which can only occur if the overall energies of the two structures are very similar. Therefore, the differences in these two structures must come from: Hydrogen Bonding, Bonding Energy, and Lattice Energy. An exploration of these variables is discussed below along with a Crystal Stress Analysis and Stress Summary in an attempt to explain why this unusual structural change occurs.

##### Hydrogen Bonding

In all three structure types, the two py-H cations H-bond to Cl ligands with average N-H•••Cl distances tabulated in Table 2. From these data, it is apparent that these distances are slightly shorter for the **Ln-6** (av 3.18 Å) in comparison to the **Ln-5** (av 3.25 Å) structures. Thus, the H-bonding of the py-H cations to Cl ligands is favored in the **Ln-6** structure by ~5 Kcal/mole [12]. Furthermore, in both structures, nearly every C-H bond of the pyridine rings forms a C-H•••Cl H-bond (see Appendix A). In the **Ln-6** structures, the two py-H cations form 13 H-bonds while in the **Ln-5** structure only 11 were observed. Thus, H-bonding is favored in the **Ln-6** structures, but by only by a slight amount since the C-H•••Cl hydrogen bonds are quite weak. In addition, the **Ln-6** structure contains a py-H-py cation. As can be seen in Table 2, this cation contains one very short and strong H-bond estimated to be ~10 Kcal/mole. In addition, each py-H-py cation forms eight or nine C-H•••Cl H-bonds, which is estimated to add another 10 Kcal/mole stabilization to the **Ln-6** structure. Summation of these values (see Appendix A) favors the **Ln-6** structure by ~25 Kcal/mole. The Cambridge Structural Database lists 125 structures with a LnCl_6_ anion [10] with 123 of these using either an ammonium, phosphonium, or protonated aromatic amine cations. Moreover, all these feature extensive H-bonding between the LnCl_6_ anions and the various cations, often involving C-H•••Cl H-bonds. This is consistent with the **Ln-6** structures reported in this work.

##### Bonding Energy

The covalency of the bonds for LnCl_6_ anions was previously investigated both experimentally and theoretically [13]. From this study, it was determined that the degree of covalent nature was between 3% and 12%. Since this contribution is so small, full ionic character was assumed and binding energies were calculated using Coulombs Law (see Appendix A for final values). For this evaluation, three energy terms were considered: (i) the Ln-Cl attraction, (ii) *cis* Cl---Cl repulsion, and (iii) *trans* Cl---Cl repulsion. Even though the Ln-Cl attraction is greater than the *cis* Cl---Cl repulsion, it was found that the **Ln-6** anion is inherently unstable when the *trans* Cl---Cl repulsion is included. In contrast, the **Ln-5** anion has less Cl---Cl repulsion due to the presence of the py ligand and is more stable than the **Ln-6** structure by a substantial amount.

If a covalent contribution is considered, the negative charge on the Cl ligands would be reduced, producing a weaker Ln-Cl attraction. This change is offset (wholly or partially) by the covalent contribution to the bond. The reduced Cl negative charge also reduces the Cl---Cl repulsions. Using an average 5% covalent contribution [13], the unfavorable GdCl_6_^−3^ anion’s energy is reduced to ~90 kcal/mol. In comparison, the LnCl_5_^−2^ becomes even more stable, as the Cl---Cl repulsion is reduced. These calculations ignore the distance controlled, ion–dipole Ln-py interaction. The Ln-N_py_ is significantly shorter than the *cis* or *trans* Cl---N distances, suggesting this has greater influence on the final stability than any Cl---N repulsion. Thus, it is assumed the Ln-N_py_ will only add to the stability of the final anion for the **Ln-5** complexes. This raises the question as to why the early Ln cations did not form the same structure. Therefore, the **Ln-6** compositions must possess a significant contribution from another source in order to generate a stable structure.

##### Lattice Energy

The final factor in the overall stability of the **Ln-6** and **Ln-5** structures would be the lattice energy. Lattice energy (U) can be written as (Equation (2)):**U = C/[R(*n*Z^+^****)(*****m*****Z^−^)]**(2)

**C** = constant, crystal structure dependent,

**R** = cation, anion distance,

**Z^+^**, **Z^−^** = cation, anion charge,

*n*, *m* = number of cations, anions.

For a 1:1 salt such as CsI, *n*, *m*, **Z^+^**, and **Z^−^** = 1, then C/R is equal to U which in this case is −144 Kcal/mole. The value of C/R that is necessary to neutralize the unfavorable energy of the NdCl_6_^−3^ anion can be calculated (C/R [(3)(+1)(1)(−3)] = 116) as C/R = −12.9 Kcal/mole. This value is very small. Thus, any cation should produce enough lattice energy to stabilize the LnCl_6_^−3^. This is evidenced by the structures reported in the CSD, where a number of LnCl_6_^−3^ anions with three monovalent cations are available. Furthermore, double salts, where the additional ions increase the lattice energy, countering the instability of the LnCl_6_ anions ([Me_2_NH_2_]_4_[LnCl_6_]Cl (Ln = Ho, Er, Tm [14]; Ln = Nd, Sm, Eu [15]) and [MeNH_3_]_4_[YbCl_6_]Cl) [16,17] have also been reported. Further examples reveal cations with higher charges, such as was noted for [Nd(dimethylurea)_6_NdCl_6_][18] or even more complex arrays such as [MeNH_3_]_8_[NdCl_6_][NdCl_4_(H_2_O)_2_]_2_Cl_3_, [19] [Me_3_PyH]_10_[ErCl_6_][ErCl_5_(H_2_O)]Cl_3_ [20] and [Nd(EO_4_)_2_]_4_[NdCl_6_]Cl_9_ (where EO_4_ = tetraethylene glycol) [21].

The C/R term contains two variables: (i) R, which is the distance between the anion and cation and (ii) the Madelung constant, which depends on the detailed arrangement of anions and cations in the lattice. Since R does not vary much between the **Ln-6** and **Ln-5** structures, it will have minimal impact on the final lattice energy. The Madelung constants for several organic salts [22] have been reported to span a small range from 1.16 to 2.52. Since the structures of both **Ln-6** and **Ln-5** are basically a shell of py-H cations surrounding a Ln-Cl complex, the Madelung constants for the **Ln-6** and **Ln-5** structures are also expected to be very similar and, to enable investigation, were assumed to be identical. Therefore, the lattice energy difference is defined by the (*n*Z^+^)(*m*Z^−^) term, which becomes 9 for **Ln-6** and 4 for **Ln-5**. At the structural switchover, the stability advantage of **Tb-5** (−283 Kcal/mol) over **Gd-6** (+88 Kcal/mol), including the binding energy of the complex anion and H-bonding is calculated to be −371 Kcal/mole (see Appendix A). Substituting for the lattice energies of **Ln-6** and **Ln-5**, the equation becomes C/R(9) = C/R(4) −371 Kcal/mol, which solves to C/R = −74 Kcal/mole. Lattice energies larger than this value will favor the **Ln-6** structure over the **Ln-5** arrangement. Recalling that the lattice energy for CsI is −144 Kcal/mole, −74 Kcal/mole is a very small lattice energy. Therefore, the **Ln-5** structure can exist only if the lattice energy is small. This conclusion was based on the assumption that the C/R term for the **Ln-6** and **Ln-5** structures are the same, which is not true. However, the range of Madelung constants is small, so even if a generous 50% difference exists between the two C/R terms, the lattice energy required for the **Ln-6** structure to dominate [−74 + 0.5(−74) Kcal/mol = −111 Kcal/mol], remains small.

##### Crystal Stress Analysis

The shortest contacts between a cationic N and a Cl ligand are listed in Appendix A. As the series is traversed the metal radii shrinks which results in a decrease in the N•••Cl contacts, resulting in slightly larger lattice energies for both the Ln-6 and Ln-5 structures. For the **Ln-6** structures, nearly all contacts stay constant or decrease slightly from La to Gd. There is, however, an anomaly as the N(70)---Cl(24) and N(70)---Cl(26) distances increase between Eu and Gd. This suggests a stress is building in the **Ln-6** structure, which is relieved after Gd due to the **Ln-5** structure transition. In the **Ln-6** structures there are four repulsive contacts between cations and all of these involve the py-H-py cation. (See Appendix A). Three of these cation–cation distances are between 3.6 and 3.8 Å and do not change across the series; however, for the fourth contact, the decrease from La (3.66 Å) to Pr (3.49 Å) to Gd (3.46 Å) results in an increasingly destabilized **Ln-6** structure. There are no such short cation–cation contacts in the **Ln-5** structure. As noted previously, the complex Cl anions in all three structures are surrounded by a shell of py-H and/or py-H-py cations. As the Ln ions become smaller, the anions shrink until there is no space in the surrounding cation layer for one of the pyridine rings. At that point the py-H-py cation is lost, tipping the stability to the **Ln-5** structure. Based upon the structures observed, this change occurs between Gd and Tb.

##### Stress Summary

From these evaluations, it is apparent that the overall energy variation between **Ln-6** and **Ln-5** is very small. The **Ln-6** structure is favored based on the lattice energy and H-bonding, whereas **Ln-5** is favored only by the greater binding energy of the LnCl_5_(py)^−2^ anion. In order for the **Ln-6** and **Ln-5** structures to co-exist, the C/R value has been estimated to be the small value of −74 Kcal/mole (*vide supra)*. Thus, the **Ln-5** structure can only exist when the lattice energy is small.

## 3. Materials and Methods

All complexes described below were handled under an atmosphere of argon with rigorous exclusion of air and water using standard Schlenk line and glove box techniques unless otherwise stated. The **[Ln(H_2_O)*_n_*(Cl)_3_]** precursors were either purchased (Sigma-Aldrich Chemical Company) or synthesized and crystallographically verified from Ln(0) or Ln_2_O_3_ in concentrated HCl. Pyridine (99.99% anhydrous) was used as received (Sigma-Aldrich).

### 3.1. Ln-6 General Synthesis

In an argon filled glovebox, **[Ln(H_2_O)*_n_*(Cl)_3_]** (1.0 g; Ln = Ce, Pr, Nd, Sm, Eu, and Gd) was mixed with pyridine (~10 mL) and stirred overnight. The resulting white slurry was heated to boiling, filtered hot through a glass frit, and the clear mother liquor allowed to slowly cool to glovebox temperature. Crystals typically grew overnight and were used for all analyses. Yields were not optimized.

**[[py-H-py][py-H]_2_[LaCl_6_]] (La-6)**. Used [La(m-Cl)(H_2_O)_7_]_2_•4Cl (1.0 g, 2.8 mmol). FTIR (KBr, cm^−1^) 3772.96(w), 3695.98(w), 3491(w), 3234.92(w), 3165.27(w), 3065.51(w), 2858.12(w), 2247.39(w), 2030.86(w), 1903.07(w), 1635.05(s), 1610.81(s), 1537.41(s), 1482.62(s), 1384.56(m), 1330.47(m), 1239.42(m), 1196.64(m), 1163.31(m), 1078.45(w), 1045.24(m), 1027.68(m), 883.5(m), 800.42(w), 742.43(s), 674.6(s), 608.56(m), 466.31(w).

**[[py-H-py][py-H]_2_[CeCl_6_]] (Ce-6)**. Used [Ce(m-Cl)(H_2_O)_7_]_2_•4Cl (1.0 g, 2.8 mmol). FTIR (KBr, cm^−1^) 3774.08(w), 3696.98(w), 3640.17(w), 3388.9(m), 3225.73(m), 3160.73(m), 3062.61(m), 2960.36(m), 2859.76(m), 2431.3(m), 2010.37(m), 1725.23(w), 1709.64(w), 1632.98(s), 1610.05(s), 1530.13(s), 1484.03(s), 1445.41(m), 1407.12(m), 1332.49(m), 1248.52(m), 1196.69(m), 1164.75(m), 1039.43(m), 1006.05(m), 895.41(m), 803.22(w), 747.48(s), 677.03(s), 630.23(m), 608.69(m), 572.35(m), 516.64(m), 503.25(m), 489.23(m), 476.23(m), 464.99(m), 432.34(w).

**[[py-H-py][py-H]_2_[PrCl_6_]] (Pr-6)**. Used [Pr(m-Cl)(H_2_O)_7_]_2_•4Cl (1.0 g, 2.8 mmol). 3910.89(w), 3860.01(w), 3845.5(w), 3027.46(w), 3774.15(w), 3696.91(w), 3682.41(w), 3661.92(w), 3638.78(w), 3230.83(m), 3162.37(m), 3066.82(m), 2958.45(m), 2918.67(m), 1902.17(m), 1725.39(w), 1709.57(w), 1690.34(w), 1660.5(w), 1634.75(s), 1610.53(s), 1524.72(s), 1482.39(s), 1447.93(w), 1408.65(w), 1384.25(w), 1330.01(m), 1238.96(m), 1196.07(m), 1163.44(m), 1045.53(m), 1027.31(m), 881.46(m), 802.8(m), 741.4(s), 672.87(s), 608.65(m), 477.2(m), 462.71(m).

**[[py-H-py][py-H]_2_[NdCl_6_]] (Nd-6)**. Used [NdCl_2_(H_2_O)_6_]•Cl (1.0 g, 2.7 mmol). FTIR (KBr, cm^−1^) 3773.75(w), 3696.26(w), 3661.34(w), 3487.35(w), 3230.64(m), 3163.07(m), 3066.45(m), 2958.88(w), 2876.61(m), 2385.43(w), 2247.28(w), 2031.26(m), 1857.95(m), 1725.2(w), 1690.23(w), 1634.91(s), 1610.21(s), 1524.15(s), 1482.68(s), 1386.38(m), 1329.89(m), 1237.87(m), 1196.17(m), 1163.32(m), 1046.5(m), 1027.85(m), 874.62(m), 802.67(m), 763.56(m), 741.07(s), 672.82(s), 608.78(m), 481.28(m), 465.46(m), 452.45(m).

**[[py-H-py][py-H]_2_[SmCl_6_]] (Sm-6)**. Used [SmCl_2_(H_2_O)_6_]•Cl (1.0 g, 2.7 mmol). FTIR (KBr, cm^−1^) 3768.91(w), 3696.9(w), 3489.46(w), 3257.29(m), 3068.2(w), 2851.65(w), 2438.27(w), 2245.76(w), 2004.26(m), 1861.29(m), 1690.42(w), 1635.38(s), 1610.82(s), 1535.49(s), 1483.38(s), 1385.18(m), 1329.81(s), 1240.08(s), 1195.9(s), 1162.83(s), 1047.83(s), 1027.69(s), 990.16(s), 879.18(s), 743.46(s), 670.09(s), 608.73(s), 477.95(w), 462.07(w).

**[[py-H-py][py-H]_2_[EuCl_6_]] (Eu-6)**. Used [EuCl_2_(H_2_O)_6_]•Cl (1.0 g, 2.7 mmol). FTIR (KBr, cm^−1^) 3846.12(w), 3772.94(w), 3696.39(w), 3662.6(w), 3639.83(w), 3471.14(w), 3234.01(m), 3169.04(m), 3070.65(m), 2898.95(m), 2594.46(w), 2469.78(m), 2348.75(w), 2244.3(m), 2005.72(m), 1969.08(m), 1857.52(m), 1726.58(m), 1709.94(w), 1690.94(m), 1660.11(m), 1634.52(s), 1610.52(s), 1534.22(s), 1484.72(s), 1444.93(s), 1404.77(m), 1385(m), 1325.69(m), 1236.86(m), 1221.76(s), 1196.81(s), 1163.36(s), 1063.65(s), 1039.57(s), 1027.82(s), 1006.78(s), 980.52(m), 870.47(s), 807.83(w), 763.36(s), 740.45(s), 710.9(m), 671.49(s), 625.68(m), 608.96(m), 477.88(w), 464(w), 441.05(s).

**[[py-H-py][py-H]_2_[GdCl_6_]] (Gd-6)**. Used [GdCl_2_(H_2_O)_6_]•Cl (1.0 g, 2.7 mmol). FTIR (KBr, cm^−1^) 3965(w), 3765.31(w), 3693.93(w), 3432.15(w), 3255.08(w), 2851.21(w), 2647.12(w), 2594.53(w), 2469.36(w), 2244.28(w), 2006.43(m), 1967.82(m), 1858(m), 1727.55(w), 1690.39(w), 1635.32(s), 1611.06(s), 1574.51(m), 1534.76(s), 1485.87(s), 1444.45(s), 1404.79(m), 1384.71(m), 1325.22(s), 1236.66(m), 1222.16(s), 1196.74(s), 1163.04(s), 1081.02(m), 1064.56(s), 1040.55(s), 1008.45(m), 979.04(s), 927.64(w), 869.32(s), 806.16(m), 762.97(s), 709.47(m), 670.24(m), 626.94(s), 608.71(m), 480.92(m), 426.02(s).

### 3.2. Ln-5 General Syntheses

In a glovebox under argon, **[Ln(H_2_O)_n_(Cl)_3_]** (1.0 g; Ln = Tb, Dy, Ho, Er, Tm, Yb, and Lu) was mixed with pyridine (~10 mL) and stirred overnight. The resulting clear solution was heated to a boil, which resulted in a slurry. The reaction mixture was filtered hot and upon cooling, X-ray quality crystals were isolated. Yields were not optimized.

**[[py-H]_2_[TbCl_5_(py)]] (Tb-5)**. Used [TbCl_2_(H_2_O)_6_]•Cl (1.0 g, 2.7 mmol). FTIR (KBr, cm^−1^) 3684.54(w), 3232.2(m), 3171.03(m), 3103.73(m), 3066.41(m), 3037.55(m), 2997.84(m), 2923.11(m), 2637.73(w), 2601.67(w), 2461.5(w), 2378.31(w), 2306.69(w), 2221.59(w), 2005.99(m), 1932.71(m), 1876.56(m), 1771.47(w), 1710.68(w), 1633.71(s), 1599.96(s), 1535.09(m), 1487.84(s), 1443.18(s), 1388.06(m), 1365.53(m), 1325.19(m), 1222.3(s), 1198.45(m), 1153.67(s), 1071.68(s), 1037.26(s), 1005.41(s), 870.42(m), 758.26(s), 698.16(s), 673.61(s), 623.03(s), 475.44(w).

**[[py-H]_2_[DyCl_5_(py)]] (Dy-5)**. Used [DyCl_2_(H_2_O)_6_]•Cl (1.0 g, 2.7 mmol). FTIR (KBr, cm^−1^) 3457.31(w), 3233.87(w), 3172.64(w), 3069.04(w), 2997.42(w), 2923.04(w), 2461.51(w), 2306.36(w), 2005.84(m), 1933.22(m), 1875.68(m), 1635.09(s), 1600.9(s), 1533.44(s), 1487.41(s), 1443.96(s), 1386.96(m), 1325.73(m), 1222.58(s), 1198.56(m), 1153.48(s), 1070.31(s), 1037.99(s), 1006.31(s), 870.74(m), 758.4(s), 701.5(s), 674.32(s), 624.16(s), 532.06(w), 477.56(w).

**[[py-H]_2_[HoCl_5_(py)]] (Ho-5)**. Used [HoCl_2_(H_2_O)_6_]•Cl (1.0 g, 2.6 mmol). FTIR (KBr, cm^−1^) 3684.47(w), 3233.83(w), 3183(w), 3070.02(w), 2922.56(w), 2638.54(w), 2461.47(w), 2380.21(w), 2306.64(w), 2006.69(w), 1933.86(w), 1875.23(m), 1776.36(w), 1711.28(w), 1690.31(w), 1634.4(s), 1600.45(s), 1534.57(s), 1487.51(s), 1442.53(s), 1387.02(m), 1366.62(m), 1324.9(m), 1222.89(s), 1197.84(s), 1154.64(s), 1071.29(s), 1037.65(s), 1005.86(s), 869.48(s), 762.16(s), 672.64(m), 623.34(m).

**[[py-H]_2_[ErCl_5_(py)]] (Er-5)**. Used [ErCl_2_(H_2_O)_6_]•Cl (1.0 g, 2.6 mmol). FTIR (KBr, cm^−1^) 3233.22(w), 3066.09(w), 2998.2(w), 2923.08(w), 2460.91(w), 2306.48(w), 2004.86(m), 1934.58(m), 1876.39(m), 1711.38(w), 1634.6(s), 1600.46(s), 1534.97(s), 1487.79(s), 1442.82(s), 1388.13(m), 1365.46(m), 1325.4(m), 1223.11(s), 1198.74(m), 1153.95(s), 1071.44(s), 1037.89(s), 1006.09(s), 949.9(w), 880.65(m), 758.62(s), 707.06(s), 623.45(s), 499.35(w), 466.09(w).

**[[py-H]_2_[TmCl_5_(py)]] (Tm-5)**. Used [TmCl_2_(H_2_O)_6_]•Cl (1.0 g, 2.6 mmol). FTIR (KBr, cm^−1^) 3454.91(w), 3233.43(m), 3172.46(m), 3104.34(m), 3070.69(m), 2997.9(w), 2962.49(w), 2922.94(m), 2461.36(m), 2224.92(w), 2006.04(m), 1874.5(m), 1710.29(w), 1635.65(s), 1601.84(s), 1535.29(s), 1487.26(s), 1444.61(s), 1387.05(m), 1325.53(m), 1223.15(s), 1198.49(m), 1154.7(m), 1070.93(s), 1038.83(s), 1006.54(s), 871.43(m), 757.97(s), 699.9(s), 674.09(s), 625.04(s), 476,7(w).

**[[py-H]_2_[YbCl_5_(py)]] (Yb-5)**. Used [YbCl_2_(H_2_O)_6_]•Cl (1.0 g, 2.6 mmol). FTIR (KBr, cm^−1^) 3468.16(w), 3233.69(w), 3172.85(w), 3070.81(m), 2923.74(w), 2009.28(w), 1864.61(w), 1636.39(s), 1602.99(s), 1535.4(s), 1486.5(s), 1444.66(s), 1325.91(m), 1222.14(s), 1198.56(m), 1163.6(m), 1066.91(s), 1039.86(s), 1007.38(s), 872.12(m), 741.21(s), 700.86(s), 674.28(s), 628.17(s), 577.87(m).

**[[py-H]_2_[LuCl_5_(py)]] (Lu-5)**. Used [LuCl_2_(H_2_O)_6_]•Cl (1.0 g, 2.6 mmol). FTIR (KBr, cm^−1^) 3233.9(w), 2923.47(w), 2007.68(m), 1859.56(m), 1636.09(s), 1610.98(s), 1534.48(s), 1485.69(s), 1445.22(s), 1405.95(w), 1385.32(m), 1324.82(s), 1224.47(s), 1197.48(s), 1163.6(s), 1065.22(s), 1041.57(s), 1028.73(s), 1008.19(s), 980.85(s), 870.5(s), 765.75(s), 740.12(s), 710.89(m), 672.71(s), 627.69(s), 608.66(m), 452.65(w).

## 4. Analytical Analyses

All samples used for analytical analyses were dried and handled under an argon atmosphere.

### 4.1. Infrared Spectroscopy

All samples were prepared under an argon atmosphere using a hand press. FT-IR spectroscopic data were collected on a Nicolet 6700 FT-IR spectrometer using a KBr pellet press under a flowing atmosphere of nitrogen.

### 4.2. X-ray Crystal Structure Information

For each sample, single crystals were mounted onto a loop from a pool of Fluorolube™ (Sigma-Aldrich, MO, USA) or Parabar 10312 (Hampton Research, Aliso Viejo, CA, USA) and immediately placed in a 100 K N_2_ vapor stream. X-ray intensities were measured using a Bruker APEX-II CCD diffractometer with MoKα radiation (λ = 0.71070 Å). Indexing, frame integration, and structure solutions were performed using the Bruker SHELXTL [23,24,25] software package within the Apex3 [23] and/or OLEX2 [26] suite of software. Additional information concerning the data collection and final structural solutions can be found by accessing CIF files through the Cambridge Crystallographic Database [10]. The unit cell parameters for all compounds are available in Table 1 and select metrical data available in Table 2. Specific details of the structure solution are discussed below.

The previous reports concerning the [[py-H]_2_[LnCl_5_(py)] (Ln = Eu, Er, Yb) [8] structures were solved in the Pnma space group. With our data in the Pnma model, the PyH cation was rotationally disordered by about 10 degrees in a 1:1 ratio for all of the **Ln-5** compounds. Additionally, for all the **Ln-5** compounds, the R value for the intrinsic phasing solution for the noncentric space group Pna2_1_ was about half that of the Pnma models. The final R values for the Pna2_1_ solutions were less than 2.3% in all cases, see Table 1. The results of a statistical significance F-test [27] are presented in Appendix A in the supporting information and these data clearly indicate, in all cases, that the Pna2_1_ model is favored at much more than the 99% confidence level. The descriptions of the two structures solutions are very similar; in each case the py-H cation resides between two [LnCl_5_p(py)] anions and forms a bifurcated N-H•••Cl bond. The previously reported Pnma structure for the Er derivative [8] possesses two py-H cations that are symmetry related by a mirror plane. This results in two mirror related N atoms, separated by 5.062 Å, H-bonding to the same pair of Cl ligands [Cl(1) and Cl(2) in our numbering system]. For the Pna2_1_ structure in this study, the two py-H cations are not equivalent. The N(20) atom H bonds to Cl(1) and Cl(2) as in the Pnma description, but the N(30) atom H bonds to Cl(2) and Cl(4). This results in a longer distance between the N atoms at 5.122 Å. The Pna21 crystals are racemically twinned with ratios between 0.350(8) and 0.500(9).

## 5. Summary and Conclusions

Attempts to dehydrate **[Ln(H_2_O)*_n_*(Cl)_3_]** using pyridine led to the isolation of two types of polychloride Ln-species, crystallographicaly characterized as [[py-H-py][py-H]_2_[LnCl_6_]] (**Ln-6**; Ln = La, Ce, Pr, Nd, Sm, Eu, Gd) or [[py-H]_2_[LnCl_5_(py)]] ((**Ln-5**; Ln = Tb, Dy, Ho, Er, Tm, Yb, Lu). Extensive H-bonding, between the py-H cations and the Ln-anions stabilizes the crystal. The addition of the py bonding energy appears to favor the **Ln-5** structures, whereas the introduction of the py-H-py cation adds enough lattice energy to favor the **Ln-6**. The structure change from **Ln-6** to **Ln-5**, is believed to occur due to the decreasing size of the Ln cation eliminating the space necessary for the py-H-py stabilizing counter cation. Thus, Coulombic energy, H-bonding and the Ln-contraction combine to dictate the final structural arrangements observed herein. It is apparent that this ‘simple’ system of **[Ln(H_2_O)*_n_*(Cl)_3_**] and pyridine is much more complex than anticipated.

## Figures and Tables

**Figure 1 molecules-28-00283-f001:**
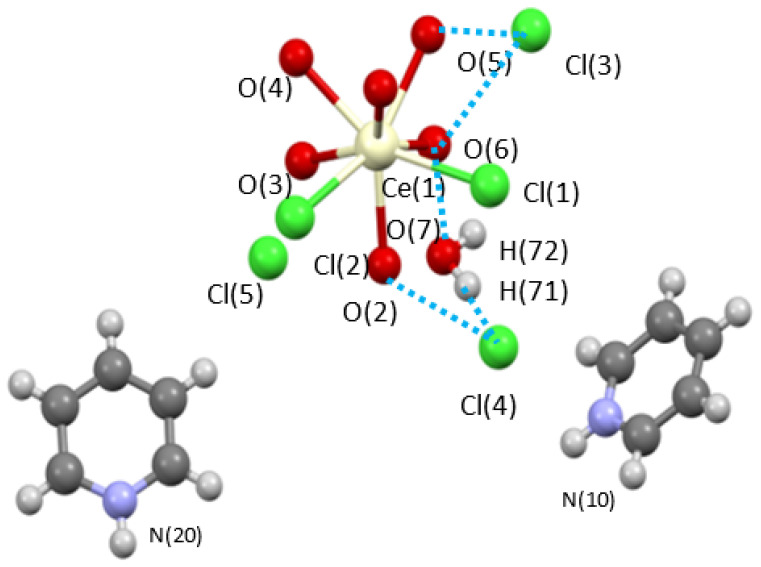
Ball and stick structure plot of [Ce(Cl)_2_(OH_2_)_6_][py-H]_2_[Cl]_3_(H_2_O) (**Ce-H_2_O/py-H**). Blue dashed line represents H-bonding.

**Figure 2 molecules-28-00283-f002:**
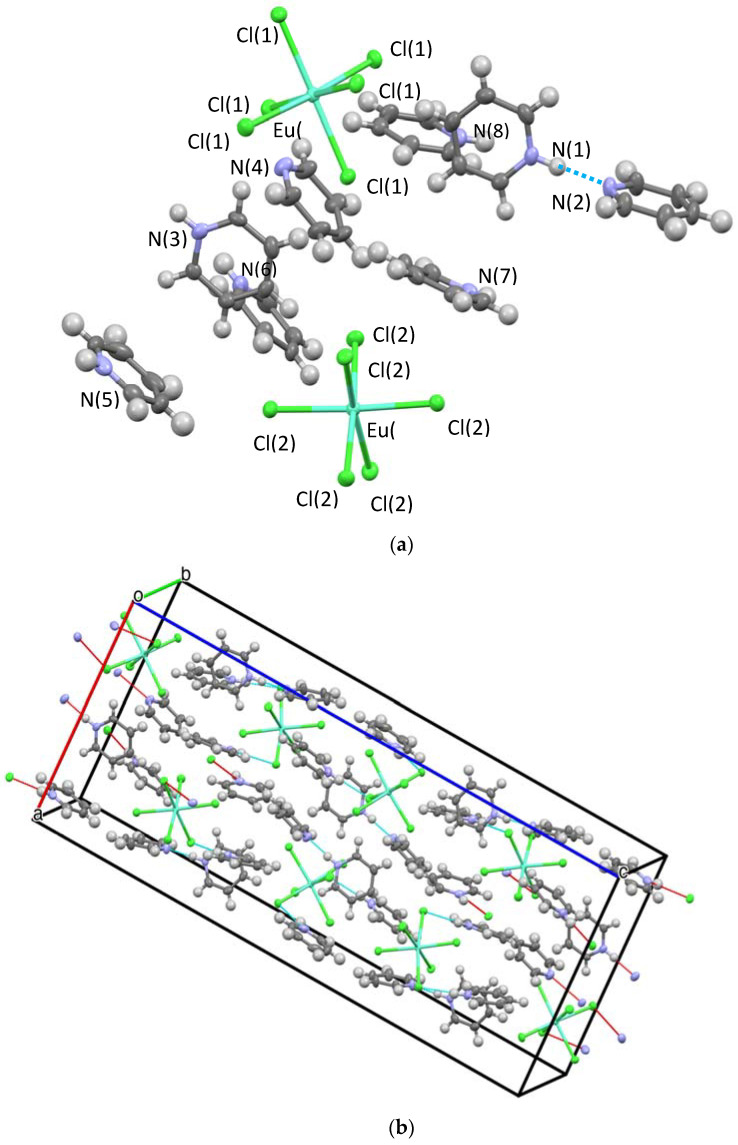
Structure plot of [[py-H-py][py-H]_2_[EuCl_6_]] (**Eu-6**) (**a**) unit cell solution and (**b**) packing diagram with H-bonding shown in blue dashes and short contacts in red. Thermal ellipsoids are drawn at 50% level.

**Figure 3 molecules-28-00283-f003:**
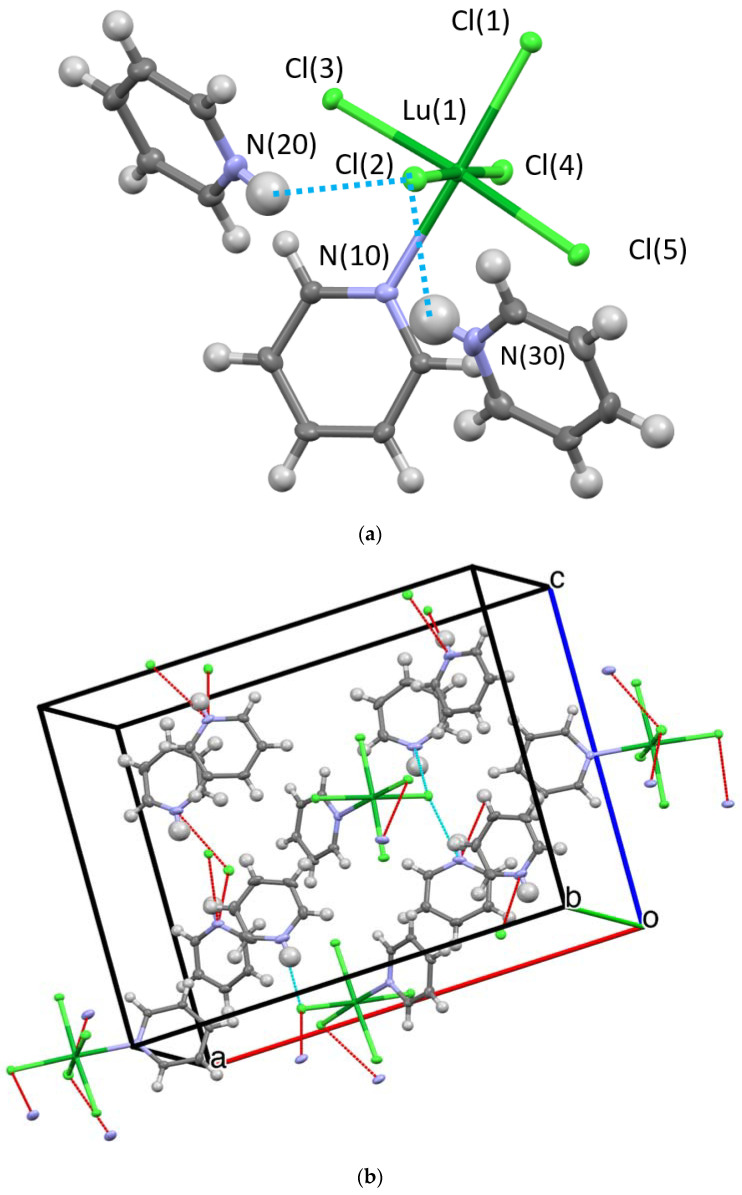
Structure plot of [[py-H]_2_[LuCl_5_(py)]] (**Lu-5**): ((**a**) unit cell solution and (**b**) packing diagram with H-bonding shown in blue dashes and short contacts in red. Thermal ellipsoids are drawn at 50% level. Blue dashed line represents H-bonding and red lines, short contacts.

**Table 1 molecules-28-00283-t001:** Data collection parameters for **La-6 to Ce-H_2_O**.

Compound	La-6	Ce-6	Pr-6
Chem. Form	C_20_H_23_Cl_6_N_4_La	C_20_H_23_CeCl_6_N_4_	C_20_H_23_Cl_6_N_4_Pr
Form. weight	335.02	672.24	673.03
temp (K)	100 (2)	100 (2)	100 (2)
space group	Orthorhombic P 2_1_ 2_1_ 2_1_	Orthorhombic P 2_1_ 2_1_ 2_1_	Monoclinic P2_1_/c
*a* (Å)	9.7104(6)	9.6382(3)	15.6181(11)
*b* (Å)	15.6614(10	15.6027(4)	9.5868(8)
*c* (Å)	18.1015(12)	18.0596(6)	36.244(3)
b (deg)			89.999(3)
V (Å^3^)	2752.8(3)	2715.84(14)	5426.8(7)
Z	4	4	8
D_calcd_ (Mg/m^3^)	1.619	1.644	1.648
m (Mo, Ka) (mm^−1^)	2.149	2.282	2.402
Flack Parameter	0.500(6)	0.50(1)	NA ^c^
R1 ^a^ (%) (all data)	2.25 (2.65)	2.22 (2.22)	5.55 (10.16)
wR2 ^b^ (%) (all data)	4.86 (5.18)	6.14 (6.15)	10.15 (12.71)
**Compound**	**Nd-6**	**Sm-6**	**Eu-6**
Chem. Form	C_20_H_23_Cl_6_N_4_Nd	C_20_H_23_Cl_6_N_4_Sm	C_20_H_23_Cl_6_EuN_4_
Form. weight	676.36	682.47	684.08
temp (K)	100 (2)	100 (2)	100 (2)
space group	Monoclinic P2_1_/c	Monoclinic P2_1_/c	Monoclinic P2_1_/c
*a* (Å)	15.561(2)	15.5230(14)	15.534(3)
*b* (Å)	9.5667(15)	9.5386(9)	9.5290(19)
*c* (Å)	36.204(5)	36.265(4)	36.287(7)
b (deg)	90.306(5)	90.244(4)	90.091(7)
V (Å^3^)	5389.3(14)	5369.6(9)	5371.5(18)
Z	8	8	8
D_calcd_(Mg/m^3^)	1.667	1.688	1.692
m (Mo, Ka) (mm^−1^)	2.537	2.800	2.948
Flack Parameter	NA	NA	NA
R1 ^a^ (%) (all data)	8.14 (13.45)	4.45 (6.04)	4.81 (9.68)
wR2 ^b^ (%) (all data)	16.45 (19.59)	9.48 (10.78)	11.22 (15.11)
**Compound**	**Gd-6**	**Tb-5**	**Dy-5**
Chem. Form	C_20_H_23_Cl_6_GdN_4_	C_15_H_17_Cl_5_N_3_Tb	C_15_H_17_Cl_5_DyN_3_
Form. weight	689.37	572.48	579.06
temp (K)	100 (2)	100 (2)	100(2)
space group	Monoclinic P2_1_/c	Orthorhombic Pna2_1_	Orthorhombic Pna2_1_
*a* (Å)	15.5351(13)	15.561(2)	18.693(3)
*b* (Å)	9.5079(8)	9.5667(15)	7.3078(10)
*c* (Å)	36.279(3)	36.204(5)	14.789(2)
b (deg)	90.055(3)		
V (Å^3^)	5358.6(8)	2028.3(3)	2020.3(5)
Z	8	4	4
D_calcd_(Mg/m^3^)	1.709	1.885	1.904
m (Mo, Ka) (mm^−1^)	3.089	4.148	4.362
Flack Parameter	NA	0.500(9)	0.459(6)
R1 ^a^ (%) (all data)	6.38 (7.36)	1.78 (1.98)	1.37 (1.40)
wR2 ^b^ (%) (all data)	13.85 (14.77)	3.65 (3.71)	3.25 (3.30)
**Compound**	**Ho-5**	**Er-5**	**Tm-5**
Chem. Form	C_15_H_17_Cl_5Ho_N_3_	C_15_H_17_Cl_5_ErN_3_	C_15_H_17_Cl_5_N_3_Tm
Form. weight	581.49	583.82	585.49
temp (K)	100 (2)	100 (2)	100 (2)
space group	Orthorhombic Pna2_1_	Orthorhombic Pna2_1_	Orthorhombic Pna2_1_
*a* (Å)	18.6738(17)	18.6315(7)	18.6333(11)
*b* (Å)	7.2976(6)	7.2898(3)	7.2815(4)
*c* (Å)	14.7913(13)	14.7705(5)	14.7648(9)
V (Å^3^)	2015.7(3)	2006.13(13)	2003.3(2)
Z	4	4	4
D_calcd_(Mg/m^3^)	1.916	1.933	1.941
m (Mo, Ka) (mm^−1^)	4.590	4.852	5.098
Flack Parameter	0.465(5)	0.486(5)	0.350(8)
R1 ^a^ (%) (all data)	1.65 (1.71)	1.05 (1.06)	2.27 (2.76)
wR2 ^b^ (%) (all data)	3.78 (3.79)	2.67 (2.68)	4.92 (5.28)
**Compound**	**Yb-5**	**Lu-5**	**Ce-H_2_O/py-H**
Chem. Form	C_15_H_17_Cl_5_N_3_Yb	C_15_H_17_Cl_5_N_3_Lu	C_10_H_26_CeCl_5_N_2_O_7_
Form. weight	589.60	591.54	603.70
temp (K)	100 (2)	100 (2)	100(2)
space group	Orthorhombic Pna2_1_	Orthorhombic Pna2_1_	Monoclinic P2_1_/n
*a* (Å)	18.6135(11)	18.8935(3)	8.4460(8)
*b* (Å)	7.2804(4)	7.27030(10)	17.5550(18)
*c* (Å)	147520(9))	14.7377(3)	15.5427(18)
b(deg)			91.534(3)
V (Å^3^)	1999.1(2)	1992.25(6)	2303.7(4)
Z	4	4	4
D_calcd_(Mg/m^3^)	1.959	1.972	1.741
m (Mo, Ka) (mm^−1^)	5.349	5.628	2.585
Flack Parameter	0.457(7)	0.488(6)	
R1 ^a^ (%) (all data)	2.16 (2.45)	1.90 (2.73)	5.50 (5.73)
wR2 ^b^ (%) (all data)	4.63 (4.81)	3.41 (3.56)	18.67 (18.84)

**^a^ R1 = Σ||F_o_| − |F_c_||/S|F_o_| × 100; ^b^ wR2 = [Σ w (F_o_^2^ − F_c_^2^)^2^/Σ (w|F_o_|^2^)^2^]^1/2^ × 100;**^c^ NA **= Not applicable.**

**Table 2 molecules-28-00283-t002:** Average, select, metrical data for **Ln-6**.

Distance (Å)	La	Ce	Pr	Nd	Sm	Eu	Gd
Ln-Cl	2.796	2.758	2.737	2.724	2.702	2.689	2.676
Cl---Cl	3.954	3.900	3.871	3.852	3.821	3.803	3.784
(py)N-H-N(py)	2.678	2.678	2.672	2.674	2.664	2.678	2.676
(py)N-H---Cl	3.165	3.171	3.210	3.179	3.164	3.171	3.181
**Angles (deg)**	**La**	**Ce**	**Pr**	**Nd**	**Sm**	**Eu**	**Gd**
*trans* Cl-Ln-Cl	173.11	172.18	174.43	175.00	174.48	173.45	174.43
(py)N-H-N(py)	171.80	170.96	174.11	174.20	173.58	173.97	171.91
(py)N-H---Cl	169.63	168.77	164.87	165.67	165.67	164.88	164.47

**Table 3 molecules-28-00283-t003:** Average, select, metrical data for **Ln-5**.

Distance (Å)	Tb	Dy	Ho	Er	Tm	Yb	Lu
Ln-Cl	2.643	2.638	2.633	2.602	2.597	2.591	2.576
Cl---Cl	3.738	3.73	3.726	3.680	3.673	3.664	3.643
Ln-N(py)	2.524	2.517	2.507	2.507	2.475	2.475	2.454
(py)N-H---Cl	3.230	3.232	3.229	3.276	3.235	3.274	3.265
**Angles (deg)**	**Tb**	**Dy**	**Ho**	**Er**	**Tm**	**Yb**	**Lu**
*trans* Cl-Ln-Cl	176.105	176.37	176.37	176.615	176.71	176.805	176.775
*tans* Cl-Ln-N(py)	177.19	177.28	177.29	177.11	177.47	178.06	177.36
(py)N-H---Cl	177.19	177.28	177.29	177.11	177.47	178.06	177.36

## Data Availability

CCDC 2214067–2214080 contains the Appendix A for 1–8. These data can be obtained free of charge via from the Cambridge Crystallographic Data Centre, 12 Union Road, Cambridge CB2 1EZ, UK; fax: (+44) 1223-336-033; or e-mail: deposit@ccdc.cam.ac.uk.

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
