# Peer review of "Pyridinium Salts of Dehydrated Lanthanide Polychlorides"

_molecules, 2022, doi:10.3390/molecules28010283_

Round 1

Reviewer 1 Report

In this paper, Boyle and co-workers have presented a simple and novel methodology to dehydrate the hydrated lanthanide chloride salts using pyridine as a solvent. The dehydration in pyridine leads to Ln-6 and Ln-5 chloride compounds adopting octahedral geometries. Though lanthanide(III) halides of type Ln-6 and Ln-5 have been previously reported, the focus of this study is to develop an easy and thorough method for this purpose.  Further, the stability of these pyridinium salts is explained on the basis of hydrogen bonding, lattice energy, and crystal stress analysis. The study is interesting and provides a new way for the dehydration of hydrated lanthanide salts. Therefore, I think this will attract the attention of a broad range of audiences interested in lanthanide chemistry. The work has been competently performed with a high scientific standard, and the results have been scholarly presented. The language of the manuscript is a bit puzzling and this doesn’t read well. Therefore, I recommend the authors to recheck the language, grammar and a thorough spelling is required. The chemistry is ok. Considering these points, I recommend a minor revision of this work prior to publication.

Author Response

Thanks for the positive support in terms of the theme of the report. 

The paper has been thoroughly edited to replace any awkward phrasing that may  have been present.

Reviewer 2 Report

The manuscript by Cramer et al. presents synthesis 14 mixed ligand lanthanide complexes and determination of their X-ray crystal structures. The manuscript is written in a good English and rather well organized. However, before it can be reconsidered for publication several issues should be addressed:

1. The composition of synthesized compounds is only determined by means of single-crystal XRD analysis and FTIR spectroscopy for bulk sample. However, the neither quantitative composition of bulk sample nor the identity of single crystal and powder samples has been confirmed. Please do at least powder XRD analysis and compare the patterns with calculated ones.

2. Page 3, lines 93-106. The description of FTIR data would be much more clear if one can have a look at FTIR spectra. Please, present FTIR spectra as a figure in main text or in SI.

3. Crystal structures of Tb-5, Dy-5, Ho-5, Er-5, Tm-5, Yb-5 and Lu-5 at 100K were refined in non-centrosymmetric spacegroup Pn21a. While previously Eu, Er, Yb analogs were reported in centrosymmetric spacegroup Pnma (in ref 8) at 193K. It worth noting that according to PLATON/CHECKCif output ADDSYMM test shows 100% fit of presented structures to be in Pnma spacegroup. To confirm the correctness of Pna21 solution authors are requested to perform the refinement of their own data in both spacegroups and show the Pna21 solution gives statistically significant decrease of R-factors. To support this please perform a Hamilton test [Acta Cryst. (1965). 18, 502-510, https://doi.org/10.1107/S0365110X65001081].

4. Crystal structures of Dy-5, Er-5, Yb-5 and Lu-5 show the Flack parameter > 0.5 for twin refinement, this means wrong absolute structure for major domain. Please invert the models.

Author Response

1.The composition of synthesized compounds is only determined by means of single-crystal XRD analysis and FTIR spectroscopy for bulk sample. However, the neither quantitative composition of bulk sample nor the identity of single crystal and powder samples has been confirmed. Please do at least powder XRD analysis and compare the patterns with calculated ones.

The referee is correct and this was pointed out in the text.  As requested, we have generated the pXRD patterns from the crystal structures for the Ln-6 and Ln-5 species.  These were compared to the experimental values and found to be in rough agreement.  There are numerous extraneous peaks present in the experimental data, which is associated with other organic by-products and possibly other products.  The ability to fully claim the only products isolated are the Ln-6 and Ln-5 is not possible but the characteristics of these compounds lead to this issue.  As the theme of the paper is more focused on the interesting structures noted (for several crystals), we have attempted to broach this issue as openly aas possible.  These patterns have been added to the Supporting Information and text added to the paper pointing to these data and the results.

  1. Page 3, lines 93-106. The description of FTIR data would be much more clear if one can have a look at FTIR spectra. Please, present FTIR spectra as a figure in main text or in SI.

The FTIR spectra have been added to the Supporting Information as requested and text added to the report pointing the reader to this information.

  1. Crystal structures of Tb-5, Dy-5, Ho-5, Er-5, Tm-5, Yb-5 and Lu-5 at 100K were refined in non-centrosymmetric spacegroup Pn21a. While previously Eu, Er, Yb analogs were reported in centrosymmetric spacegroup Pnma (in ref 8) at 193K. It worth noting that according to PLATON/CHECKCif output ADDSYMM test shows 100% fit of presented structures to be in Pnma spacegroup. To confirm the correctness of Pna21 solution authors are requested to perform the refinement of their own data in both spacegroups and show the Pna21 solution gives statistically significant decrease of R-factors. To support this please perform a Hamilton test [Acta Cryst. (1965). 18, 502-510, https://doi.org/10.1107/S0365110X65001081].

An F-test has been performed and all questionable structure solutions noted have been found to be in the proper space group.

  1. Crystal structures of Dy-5, Er-5, Yb-5 and Lu-5 show the Flack parameter > 0.5 for twin refinement, this means wrong absolute structure for major domain. Please invert the models.

These four structures have been rerun as requested and the tables, cif, and cifCheck in the SI have been updated for the new refinement. The updated files have been submitted to CCDC.

Round 2

Reviewer 2 Report

In revised manuscript authors adressed all reviewers' comments. From my point of view, the manuscript became clearlier. There are some issues with the purity of the compounds, since the by-products might affect the FTIR data. Nevertheless, the paper is mainly focused on structural features, therefore it can be accepted for publication after appropriate corrections made in the text.

I was somewhat surprised by the form in which the authors presented the comparison of theoretical and experimental X-ray patterns, as I expected that both curves for each compound would be presented on the same figure at an acceptable resolution.